# Assaying *Chlamydia pneumoniae* Persistence in Monocyte-Derived Macrophages Identifies Dibenzocyclooctadiene Lignans as Phenotypic Switchers

**DOI:** 10.3390/molecules25020294

**Published:** 2020-01-11

**Authors:** Eveliina Taavitsainen, Maarit Kortesoja, Tanja Bruun, Niklas G. Johansson, Leena Hanski

**Affiliations:** 1Drug Research Program, Division of Pharmaceutical Biosciences, Faculty of Pharmacy, University of Helsinki, P.O. Box 56, FI-00014 Helsinki, Finland; eveliina.taavitsainen@helsinki.fi (E.T.); maarit.kortesoja@helsinki.fi (M.K.); 2Drug Research Program, Division of Pharmaceutical Chemistry and Technology, Faculty of Pharmacy, University of Helsinki, P.O. Box 56, FI-00014 Helsinki, Finland; tanja.bruun@helsinki.fi (T.B.); niklas.johansson@helsinki.fi (N.G.J.); 3Drug Research Program, Division of Pharmacology and Pharmacotherapy, Faculty of Pharmacy, University of Helsinki, P.O. Box 56, FI-00014 Helsinki, Finland

**Keywords:** bacterial persistence, dormancy, glutathione, antibacterial agent, adjuvant therapy, natural product

## Abstract

Antibiotic-tolerant persister bacteria involve frequent treatment failures, relapsing infections and the need for extended antibiotic treatment. The virulence of an intracellular human pathogen *C. pneumoniae* is tightly linked to its propensity for persistence and means for its chemosensitization are urgently needed. In the current work, persistence of *C. pneumoniae* clinical isolate CV6 was studied in THP-1 macrophages using quantitative PCR and quantitative culture. A dibenzocyclooctadiene lignan schisandrin reverted *C. pneumoniae* persistence and promoted productive infection. The concomitant administration of schisandrin and azithromycin resulted in significantly improved bacterial eradication compared to sole azithromycin treatment. In addition, the closely related lignan schisandrin C was superior to azithromycin in eradicating the *C. pneumoniae* infection from the macrophages. The observed chemosensitization of *C. pneumoniae* was associated with the suppression of cellular glutathione pools by the lignans, implying to a previously unknown aspect of chlamydia–host interactions. These data indicate that schisandrin lignans induce a phenotypic switch in *C. pneumoniae*, promoting the productive and antibiotic-susceptible phenotype instead of persistence. By this means, these medicinal plant -derived compounds show potential as adjuvant therapies for intracellular bacteria resuscitation.

## 1. Introduction

In the course of evolution, bacteria have developed various means for protecting themselves from unfavorable conditions. Described as a reversible dormant phenotype, persistence has been acknowledged as one major survival strategy of bacteria [1,2,3]. Bacterial persistence is considered a major cause of antibiotic treatment failures and relapsing infections and it also contributes to the rise of antibiotic resistance [4]. Owing to redundant mechanisms, these phenotypical variants are able to survive under antibiotic pressure and revert back to metabolically more active phenotype when stressful conditions are cleared off.

In clinical settings, bacterial dormancy is associated with hard-to-treat infections via two mechanistically overlapping phenomena; persistent infections evading host immune responses and antibiotic persistence defined based on the presence of drug-tolerant subpopulations of bacteria. Both of these features are typical to infections caused by *Chlamydia pneumoniae*, a gram-negative obligate intracellular human pathogen that causes respiratory infections from dry cough to pneumonia. While a majority of *C. pneumoniae* infections are subclinical, with nearly everyone getting infected during their lifetime, the bacterium is also responsible for 5%–10% of community-acquired pneumonia cases worldwide [5,6]. *C. pneumoniae* has a unique biphasic development cycle, where the bacteria switch between an infectious form elementary body (EB) and a non-infectious metabolically active reticulate body (RB) [7]. The acute phase may also be followed by a persistent infection [8], occurring spontaneously in monocytes and macrophages [6,9]. A morphological hallmark of the persistent phenotype is the emergence of abnormal RBs with low metabolic activity and replication [8,10]. This viable but non-cultivable phenotype of *C. pneumoniae* is also characterized by the impairment of transcription and translation as well as the cessation of infectious EB progeny production [11].

Besides the acute respiratory illnesses, *C. pneumoniae* has been related to many chronic inflammatory diseases, such as atherosclerosis and asthma exacerbation [12,13]. The ability of *C. pneumoniae* to persist in infected cell populations forms the basis for the hypotheses on these disease connections, and monocytes and macrophages have a main role in the initiation of the chlamydial persistence [6,8].

Owing to its propensity for persistence, *C. pneumoniae* is a challenging target for antibacterial therapy. The bacterium can be eradicated from permissive cell lines such as the airway epithelium by macrolide and tetracycline antibiotics [14] but its complete eradication remains challenging. New strategies for combatting the persistent chlamydial populations are thus needed. Within our earlier work, we have identified the antichlamydial activity of dibenzocyclooctadiene lignans isolated from a medicinal plant *Schisandra chinensis* on *C. pneumoniae* replication in respiratory epithelial cells [15,16]. Based on the selectivity studies on other gram-negative and gram-positive bacteria as well as by the lack of published reports on antibacterial effects of these lignans, they seem to exert a narrow spectrum of antibacterial activity limited to *Chlamydia* spp. bacteria and possibly some other intracellular pathogens. Rather than as antimicrobial agents, the *S. chinensis*-derived lignans have been widely studied on their histoprotective effects on liver, heart, kidney, and central nervous tissues [17,18,19,20].

To date, a variety of anti-persister molecules have been described against both gram-positive and gram-negative human pathogens, major strategies involving direct eradication of the metabolically quiescent cells by e.g., membrane-active compounds, bacterial resuscitation by boosting energy metabolism and application of combination therapies [4]. According to current consensus, taking persister bacteria into account is a critical success factor in antibacterial therapy. Regarding *C. pneumoniae*, only a few membrane-active agents capable of affecting EB infectivity independent of metabolic activity have been described [21,22] and to date, no agents capable to eradicate the persistent intracellular forms of the bacterium have been described. Furthermore, current standard methods for antichlamydial susceptibility testing are based solely on permissive epithelial cells and involve the suppression of host cell responses by cycloheximide treatment [23,24].

To identify antichlamydial agents yielding improved eradication of *C. pneumoniae* also in the persistence prone, nonpermissive host cell populations, we studied *C. pneumoniae* infection in THP-1 macrophages by quantitative PCR and quantitative culture. With this model, we identified the differential antichlamydial potency of schisandrin and schisandrin C on *C. pneumoniae* persistence. These dibenzocyclooctadiene lignans show activities that are qualitatively and quantitatively different from each other and the standard of care antibiotic azithromycin. While schisandrin promoted the productive *C. pneumoniae* infection, schisandrin C yielded a highly effective eradication of the infection. Furthermore, combining schisandrin to azithromycin treatment resulted in significantly improved antibacterial effect than sole azithromycin treatment, indicating the potential of this approach as an adjuvant therapy for current antibiotics.

## 2. Results

### 2.1. Schisandrin Lignans as Modulators of C. pneumoniae Infection

Encouraged by our earlier findings on the antichlamydial activity of dibenzocyclooctadiene lignans against the productive, actively replicating *Chlamydia* spp. bacteria [15,16], three of these lignans, schisandrin, schisandrin B, and schisandrin C (Figure 1) were evaluated for their efficacy against *C. pneumoniae* in macrophages.

To define lignan concentrations tolerated by the macrophages, THP-1 cell viability upon lignan exposure was determined with two assays, the resazurin reduction assay and the intracellular ATP quantification (Table 1). Schisandrin and schisandrin C were well tolerated after 24–144 h exposure, but schisandrin B concentrations above 10 µM decreased THP-1 macrophage viability. Since the *C. pneumoniae* infection caused a small but significant (24 h, *P* = 0.002; 48 h, *P* = 0.04; 72 h, *P* = 0.043) increase in ATP levels the quantification of intracellular ATP was also performed in the presence of *C. pneumoniae* infection, yielding viability values essentially similar to those observed in noninfected cells.

The impact of the schisandrin lignans on *C. pneumoniae* growth kinetics was determined in THP-1 macrophages by following the bacterial genome copy numbers with qPCR in various time points from 2 to 144 h post infection.

Azithromycin, the standard of care antibiotic for chlamydial infections at its typically used concentration 20 nM did not suppress the bacterial replication in the macrophages. In contrast, the same azithromycin concentration yields 99% inhibition in permissive respiratory epithelial cells (Appendix B
Figure A1). At 100 nM azithromycin reduced the *C. pneumoniae* genome numbers by 68% of infection at 72 h post infection and by 93% at 144 h post infection (Figure 2). This indicates that *C. pneumoniae* survival is significantly improved under antibiotic pressure in the macrophages compared to the epithelial cells. The prolonged antibiotic tolerance is a typical hallmark for the presence of persister subpopulations [1].

As shown in Figure 2, 10 μM schisandrin B and 25 μM schisandrin C were as effective as 100 nM azithromycin in reducing *C. pneumoniae* genome numbers at 72 h and 144 h after infection. At 50 μM, schisandrin C was superior to all other samples assayed for *C. pneumoniae* genome number reduction and showed a statistically significant difference to azithromycin in this respect. It yielded 95% reduction in bacterial genome numbers at 72 h and 99% reduction at 144 h, respectively.

The effect of the schisandrin lignans on *C. pneumoniae* infectious progeny production in THP-1 macrophages was also evaluated. Within its productive life cycle, *C. pneumoniae* infectious progeny production occurs typically 48–72 h post infection, involving differentiation of the newly formed bacterial cells into EBs that leave the host cell to infect neighboring cells. For detecting the production of infectious EB progenies in THP-1 macrophages, inoculation of infected THP-1 cell lysates and culture medium supernatants into permissive human lung (HL) cells was carried out. By subsequent immunostaining of chlamydial inclusions in the HL monolayers with chlamydia-specific anti-LPS antibody, the quantities of infectious progeny in the original samples was determined. 

As shown in Table 2, schisandrin B and schisandrin C decreased *C. pneumoniae* infectious progeny production in a statistically significant manner at 72 and 144 h. In schisandrin C treated samples (25 μM or 50 μM), not any characteristic inclusions were detected, and only some small irregular inclusion-like structures were observed in HL monolayers inoculated with the cell lysates and none in those inoculated with supernatant samples. In contrast, azithromycin failed to clear the cell lysates from *C. pneumoniae*, yet azithromycin treated culture supernatants contained no bacterial progeny. Interestingly, schisandrin increased the bacterial progeny levels in particular after 72 h post infection. At 25 μM schisandrin treatment resulted in a statistically significant increase in infectious EB quantities detected in THP-1 cell lysates at 72 h post infection.

### 2.2. Concomitant Administration of Schisandrin and Azithromycin

The impact of concomitant administration of schisandrin and azithromycin on *C. pneumoniae* survival was determined. The ability of 25 μM schisandrin treatment to increase *C. pneumoniae* bacterial progeny production indicates that this lignan concentration promotes *C. pneumoniae* productive infection, shifting the balance of the intracellular bacteria populations to favor the replicative state. The bacteriostatic activity of azithromycin is targeted to replicative bacteria, and it was thus hypothesized that *C. pneumoniae* eradication in macrophages could be improved by the simultaneous administration of the two compounds. Figure 3 presents data on experiments made to verify this. Consistent with the findings shown above, schisandrin did not affect *C. pneumoniae* total genome copy numbers and azithromycin yielded approximately 49% suppression on them at 72 h post infection (Figure 3A). Treating the infection simultaneously with both compounds yielded approximately 91% reduction in *C. pneumoniae* genome copy numbers, which is statistically highly significant (*P* value 0.00042) compared to sole azithromycin treatment. Furthermore, administering schisandrin together with azithromycin drastically increased the eradication of progeny EBs from the cultures (Figure 3B, *P* value 0.0022).

### 2.3. Role of Cellular Redox Status in the Effects of Schisandrin Lignans on C. pneumoniae Infection

As the dibenzocyclooctadiene lignans are known for their redox activities and changes in cellular redox status have also been linked to bacterial persistence, we addressed the differential effects of schisandrin and schisandrin C on *C. pneumoniae* infection by studying changes in cellular redox status. As shown in Figure 4A, both multiplicity of infection (MOI) 1 and MOI5 *C. pneumoniae* infections of THP-1 macrophages elevated the ROS production of the cells statistically significantly at 48 h post infection. After 72 h infection ROS levels were elevated in MOI5-infected samples. When THP-1 macrophages were treated with schisandrin and schisandrin C, no differences in basal ROS levels were observed after 4–48 h exposure. After 72 h schisandrin and schisandrin C elevated ROS levels at both 25 μM and 50 μM concentration (Figure 4B). The concomitant administration of schisandrin and schisandrin C with MOI5 *C. pneumoniae* infection did not change the detected ROS levels compared to a vehicle-treated infection control (Figure 4C).

The impact of *C. pneumoniae* infection and schisandrin lignans on redox status of THP-1 macrophages was further studied by determining cellular glutathione (GSH) concentrations after infection, lignan treatment, or the combination of these two. As shown in Figure 5A, *C. pneumoniae* infection caused a time—and infection MOI-dependent elevation in cellular GSH levels, detectable 48–72 h post infection.

Our previous studies in monocytic THP-1 cells showed that the lignans affect cellular glutathione metabolism by causing a decrease in total GSH pools [25]. Similar to monocytic THP-1 cells, THP-1 macrophages exhibited remarkably lowered GSH pools after lignan treatment (Figure 5B), and a drastic decrease in GSH levels was also observed in *C. pneumoniae*-infected THP-1 macrophages after lignan exposure (Figure 5C).

The relevance of GSH depletion in the lignans’ activities on *C. pneumoniae* infection phenotype was evaluated by supplementing the infected cultures with GSH ethyl ester. While administration of this cell-permeable GSH derivative alone increased EB yields by approximately 65%, supplementation of schisandrin-treated infections yielded infectious progeny levels similar or lower than those in the infection control (Figure 6). The GSH supplementation was thus sufficient to eliminate the elevating effect of the 25 μM schisandrin treatment on bacterial progeny production.

## 3. Discussion

Current medication of chlamydial infections relies mostly on macrolide and tetracycline antibiotics and recurrent infections emerging after the antibiotic therapy are commonly observed. As obvious also from our data on the failure of azithromycin to clear the *C. pneumoniae* infection from monocyte-derived macrophages, improved antichlamydial agents active also on dormant bacteria are urgently needed. While azithromycin did clear the culture supernatants from infectious bacterial progeny, it was able to only partially suppress the intracellular bacteria loads. In this respect, the efficacy of schisandrin C was superior to azithromycin, as both intracellular and extracellular bacteria were cleared from the cultures by the lignan. Furthermore, the concomitant administration of schisandrin with azithromycin significantly improved the antichlamydial efficacy compared to sole azithromycin treatment. Based on these findings as well as the observation of increased EB progeny production by schisandrin, we propose that schisandrin acts as a *C. pneumoniae* phenotypic switcher. By shifting the intracellular bacteria populations to favor the replicative rather than dormant phenotype, it improves the antibiotic efficacy of azithromycin.

To date, only a few drug-like molecules have been described as phenotypic switchers reverting persister bacteria from dormancy to active growth [4,26,27] and to our knowledge, the schisandrin lignans represent the first phenotypic switchers described to be active on intracellular bacteria. Based on rodent bioavailability studies, micromolar plasma concentrations of the lignans can be achieved after a single oral dose [28], indicating the potential of the lignans as scaffolds for orally administered drugs. Furthermore, our in vitro cell viability data (Table 2) and the numerous in vivo studies on these compounds [29] indicate the lack of acute or subacute toxicity of the lignans. It is, however, noteworthy that THP-1 macrophage viability was significantly decreased by high concentrations of schisandrin B (Table 1). In our earlier studies with epithelial cells and undifferentiated, monocytic THP-1 cells schisandrin B was well tolerated [16,25] highlighting the cell-type dependent bioactivities of these lignans.

As key players in innate immunity, macrophages respond to microbes and other danger signals by generating effector molecules such as reactive oxygen species (ROS) and nitric oxide (NO) intended to kill the pathogen [30]. While NADPH oxidase is considered the major source of ROS in stimulated macrophages, mitochondrial ROS production also takes part in macrophage responses to bacterial invaders via Toll-like receptor (TLR) signaling [31]. The significance of mitochondria in innate immune responses is supported also by the findings indicating that mitochondria are targets of active manipulation by intracellular pathogens such as *Chlamydia* spp. Rather than simply escaping innate immunity responses, *C. pneumoniae* has been suggested to exploit them to drive virulence and successful infections, illustrated by the NLRP3 inflammasome activation and its contribution to *C. pneumoniae* survival in macrophages [32].

Earlier work on anti-persister agents has indicated that promotion of oxidative stress by increased ROS production eradicates persister bacteria and enhances bacterial killing by conventional antibiotics [33,34]. The medicinal plant-derived schisandrin lignans have been reported to harbor a spectrum of biological activities, such as neuro- and cytoprotective as well as anti-inflammatory properties [29]. The pharmacology of schisandrin lignans is linked to their ability to affect cellular redox status and in particular, their modulation of mitochondrial functions [35,36]. Despite the impact of the lignans on macrophage basal ROS levels (Figure 4B), ROS promotion is not likely the primary mode of action of the compounds against *C. pneumoniae*, as the infection-induced ROS levels show no difference between treated and non-treated cells (Figure 4C). Furthermore, similar impact on basal ROS levels was observed for schisandrin and schisandrin C despite their opposite effects on *C. pneumoniae* progeny yields (Table 2).

Consistent with earlier findings [37,38], we observed time-dependent fluctuation of GSH pools after chlamydial infection, which can be considered a homeostatic response of the host to the infection-induced increase in ROS levels. Our replication results show that despite the elevated ROS levels (Figure 4A) and altering GSH levels (Figure 5A) during infection, a significant fraction of the bacteria maintains an actively replicating phenotype. Thus, in contrast to murine macrophages [39,40] the oxidative environment seems not to result in purely persistent infection phenotype in human-derived THP-1 macrophages.

We also observed a remarkable decrease of THP-1 macrophage GSH levels upon lignan treatment. While our earlier studies with monocytic THP-1 cells show glutathione pool suppression by the lignans [25], earlier work has indicated that schisandrin C may increase GSH levels in the central nervous system [41] and several related lignans may prevent oxidant-induced GSH depletion in liver [17]. Based on these findings, it seems obvious that the effects of dibenzocyclooctadiene lignans on mammalian cell glutathione metabolism are dependent on target cells or tissues as well as the physiological conditions.

The lignan-induced decrease in GSH levels and the ability of GSH supplementation to revert infectious progeny production elicited by schisandrin indicate that the lignans’ impact on *C. pneumoniae* infection may involve modulation of GSH homeostasis. The cellular GSH balance may affect intracellular bacteria by a variety of mechanisms. GSH has been found to act as a major cysteine source of intracellular bacteria [42] and it has been reported to indirectly affect chlamydial energy supply by increasing cell wall permeability [43]. GSH depletion is also known to induce K^+^ efflux [44] which, in turn, can promote the chlamydial replication via the induction of NLRP3 inflammasome in the host cells [32]. On the other hand, GSH and its metabolites are directly toxic to some intracellular bacteria [45] and the GSH-dependent changes in cellular redox status may detrimentally affect *C. pneumoniae* survival, as suggested in murine models [37]. Interestingly, GSH is also known to induce virulence gene expression of *Listeria monocytogenes* [46], implying that it may serve as an indicator of the local environment, directing virulence gene expression of intracellular bacteria. To date, processes linked to the ability of *Chlamydia* spp. bacteria to sense their microenvironment have remained poorly understood, and the potential role of GSH in chlamydial adaptation and balance between active and persistent phenotype warrants further investigation. By unraveling this aspect of chlamydial biology, the schisandrin lignans act as chemical probes providing valuable insights into pathogen–host interactions.

Despite their similar effects on cellular ROS and GSH levels, schisandrin and schisandrin C show differential activity on the *C. pneumoniae* infection in THP-1 macrophages. This may reflect two separate aspects of their biological activities: a redox-dependent phenotypic switch by *C. pneumoniae* from persister to active replication and a polypharmacological nature of antichlamydial action by schisandrin C.

Our earlier findings applying permissive epithelial cells demonstrate that schisandrin C exhibits chlamydiocidal activity in the acute infection model [16]. In contrast, schisandrin does not affect *C. pneumoniae* replication at 25 μM concentration. Based on these observations, we propose that depleting cellular GSH by both of the studied lignans stimulates chlamydial growth, and persistent infection is converted to active state. In the case of schisandrin this is seen as the promotion of infectious progeny formation. Schisandrin C, on the other hand, seems to act via two distinct mechanisms: switching bacteria from persistence to active growth and active killing of the replicative bacteria.

## 4. Materials and Methods

### 4.1. General

The purity of the lignans was determined by HPLC and were 99.0% for schisandrin, 98.0%, for schisandrin B, and 98.6% for schisandrin C. The nuclear magnetic resonance (NMR) spectra (^1^H NMR, ^13^C NMR) (available as Appendix A) of the key target compounds schisandrin and schisandrin C were recorded on a Bruker Ascend 400—Avance III HD NMR spectrometer (Bruker Corporation, Billerica, MA, USA). Chemical shifts (*δ*) are reported in parts per million (ppm) relative to the NMR solvent signals (DMSO-*d_6_* 2.50 ppm and 39.50 ppm for ^1^H and ^13^C NMR, respectively).

### 4.2. Compounds

Schisandrin was obtained from Sigma-Aldrich, St. Louis, MO, USA and schisandrin B and schisandrin C were purchased from Fine Tech Industries, London, UK. ^1^H and ^13^C NMR of schisandrin and schisandrin C showed corresponding peaks and multiplicities as reported in the literature [47]. Deviations in the chemical shifts (due to different NMR solvents and spectrometers) were not greater than 0.28 ppm (^1^H) and 1.64 ppm (^13^C) for schisandrin, and 0.11 ppm (^1^H) and 0.88 ppm (^13^C) for schisandrin C.

**Schisandrin:** White solid; ^1^H NMR (400 MHz, DMSO-*d_6_*) *δ* = 6.73 (s, 1H), 6.66 (s, 1H), 3.98 (s, 1H), 3.81 (s, 3H), 3.80 (s, 3H), 3.72 (s, 3H), 3.72 (s, 3H), 3.40 (s, 3H), 3.39 (s, 3H), 2.71 (dd, *J* = 13.8, 2.0 Hz, 1H), 2.37 (d, *J* = 13.3 Hz, 1H), 2.29 (t, *J* = 4.2 Hz, 1H), 2.25 (t, *J* = 4.1 Hz, 1H), 1.71–1.64 (m, 1H), 1.12 (s, 3H), 0.70 (d, *J* = 7.1 Hz, 3H). ^13^C NMR (101 MHz, DMSO-*d_6_*) *δ* = 151.2, 151.2, 151.0, 150.6, 139.4, 139.3, 134.5, 133.7, 122.8, 122.8, 111.2, 110.6, 70.9, 60.4, 60.3, 60.0 (2C), 55.7, 55.5, 41.1, 40.5, 34.2, 30.0, 15.6.

**Schisandrin C**: White solid; ^1^H NMR (400 MHz, DMSO-*d_6_*) *δ* = 6.57 (s, 2H), 5.99 (s, 2H), 5.99–5.98 (m, 2H), 3.72 (s, 3H), 3.70 (s, 3H), 2.54–2.48 (m, 1H*), 2.26–2.23 (m, 1H), 2.08–2.03 (m, 1H), 1.98–1.95 (m, 1H), 1.84–1.76 (m, 1H), 1.70–1.60 (m, 1H), 0.92 (d, *J* = 7.1 Hz, 3H), 0.65 (d, *J* = 7.1 Hz, 3H). ^13^C NMR (101 MHz, DMSO-*d_6_*) *δ* = 148.3, 147.2, 140.7, 140.5, 137.5, 134.2, 133.9, 132.1, 121.7, 120.6, 105.6, 102.7, 100.7, 100.7, 59.0, 59.0, 40.4, 38.0, 34.5, 32.9, 21.3, 12.2. *Overlap with solvent, verified by HSQC.

The lignans, as well as azithromycin (≥98%, Cayman Chemicals, Ann Arbor, MI, USA) used as a reference antibiotic were dissolved in dimethyl sulfoxide (DMSO) and diluted in cell culture media at indicated concentrations.

### 4.3. Cell Culture

All cell cultures were maintained at 37 °C, 5% CO_2_, and 95% air humidity. THP-1 cells (ATCC TIB-202, RRID: CVCL_0006) were maintained in RPMI 1640 Dutch edition medium (Gibco, Invitrogen, Thermo Fisher Scientific, Waltham, MA, USA) supplemented with 10% FBS (BioWhittaker, Lonza, Basel, Switzerland), 2 mM l-glutamine (BioWhittaker, Lonza, Basel, Switzerland), 0.05 mM mercaptoethanol (Gibco, Invitrogen, Thermo Fisher) and 20 µg/mL gentamicin (Sigma-Aldrich, St. Louis, MO, USA). For differentiation into macrophage-like cells, THP-1 cells were incubated for 48–72 h with 160 nM phorbol-12-myristate-13-acetate (PMA, Sigma-Aldrich, St. Louis, MO, USA). Human HL cells [48] were obtained from professor Pekka Saikku/National Health Institute and University of Oulu, Finland) and maintained in RPMI 1640 (BioWhittaker, Lonza, Basel, Switzerland) supplemented with 7.5% FBS, 2 mM l-glutamine and 20 µg/mL gentamicin. When seeding HL cells into well plates, an overnight incubation was applied prior to the experiment.

### 4.4. Infections

For qPCR and infectious progeny experiments, the cells were seeded into 24-well plates (HL cells at a density of 4 × 10^5^ cell per well, THP-1 macrophages 3.5 × 10^5^ cell per well) and infected with *C. pneumoniae* (strain CV-6, obtained from professor Matthias Maass, Paracelsus Medical University, Salzburg, Austria, propagated as previously described [49]). Cell monolayers were centrifuged at 550 *g* for 1 h and incubated 1 h in 37 °C. Then, fresh medium or medium with compounds was added and the cultures were incubated from 24 to 144 h. To determine effect of GSH on *C. pneumoniae* infection, 2 mM GSH ethyl ester was added to the infected cultures at 2, 24, or 48 h post infection. For HL cell infections, cell culture medium was supplemented with 1 µg/mL of cycloheximide (CHX, Sigma-Aldrich, St. Louis, MO, USA).

### 4.5. Quantitative PCR

DNA from cell cultures was extracted with a GeneJet Genomic DNA purification kit (Thermo Fisher Scientific, Waltham, MA, USA) according to the manufacturer’s instructions for mammalian cells. The DNA concentration in samples was measured with Multiskan Sky Microplate spectrophotometer by µDrop Plate and the DNA was stored at −20 °C until use. An established qPCR method on *C. pneumoniae* ompA gene [50] was applied to quantify *C. pneumoniae* genome copy numbers. Using Step One plus Real-Time PCR system (Thermo Fisher Scientific, Waltham, MA, USA). The primers were selected for qPCR run as follows: forward primer, VD4F (5′-TCC GCA TTG CTC AGC C-3′) and reverse primer, VD4R (5′-AAA CAA TTT GCA TGA AGT CTG AGA A-3′). The reactions were prepared in 96-well MicroAmp optical plate by adding 20 ng of extracted DNA to 10 µL of master mix to 20 µL qPRC reaction. Detection was performed with Step One plus Real-Time PCR system by using the manufacturer’s standard protocol. Conditions in thermal cycle were 95 °C for 20 s and 40 cycles of 95 °C for 3 s and 60 °C for 30 s. All qPCR experiments were run with four biological replicates, each quantified as two technical replicates.

### 4.6. Infectious Progeny Assay

The EBs were harvested from infected THP-1 macrophages at various time points (72–144 h) post infection by collecting the culture supernatants, centrifuging them at 21,000 rpm for 1 h in 4 °C and resuspending the pellets with 0.2 mL of fresh cold medium. Monolayer cells from the same samples were scraped to 0.2 mL of fresh cold medium. All samples were stored in −80 °C.

The bacterial progeny in the samples was quantified by inoculating them on HL monolayers with CHX. After infection, fresh medium with CHX was added and cells were incubated for 70 h. Then, cells were fixed and stained with a genus-specific anti-LPS antibody (Pathfinder, Bio-rad, Hercules, CA, USA). Chlamydial titers were determined based on the inclusion counts observed with fluorescence microscope. The experiments were performed with four biological replicates, each quantified as a minimum of six technical replicates.

### 4.7. Resazurin Assay

THP-1 cells were seeded into 96-well plates at density of 6 × 10^4^ cells/well and the cells were differentiated with 160 nM of PMA. After differentiation, 10, 25, and 50 µM of schisandrin, schisandrin B, and schisandrin C was added to the wells. Usnic acid (50 µM, Sigma-Aldrich, St. Louis, MO, USA) was used as positive control and 0.25% DMSO as a vehicle control. Cells were incubated for 24–144 h. Culture medium with samples was replaced with fresh aliquots at 72 h in the 144 h experiment. After incubation, resazurin (Sigma-Aldrich, St. Louis, MO, USA) in PBS was added in a final concentration of 20 µM. The culture was incubated for further 2 h and fluorescence was recorded at 570/590 nm with Varioskan Lux plate reader. The experiments were performed with three biological replicates.

### 4.8. ATP Quantification Assay

THP-1 cells were seeded into 96 well plates at density of 2 × 10^4^ cells/well. Cells were differentiated with 160 nM PMA for 72 h. Then cells were infected with MOI1 of *C. pneumoniae* and after 1 h centrifugation at 550 *g* and 1 h incubation at 37 °C cells were exposed to 10, 25, and 50 µM of schisandrin, schisandrin B and schisandrin C. After 24, 48, and 72 h incubation, 100 µL of CellTiter-Glo reagent (Promega, Madison, WI, USA) was added to the wells and the plate was incubated for 10 min at room temperature. To minimize the leakage of the signal, 100 µL of the lysate was transferred to a white 96-well plate. Luminescence was recorded with Varioskan Lux plate reader. The experiments were performed with three biological replicates.

### 4.9. Intracellular ROS Detection Assay

THP-1 macrophages in 96-well plate (6 × 10^4^ cells/well) were subjected to an infection by *C. pneumoniae*, treatment with the schisandrin lignans or a combination of these two, using a culture medium without mercaptoethanol. In experiments with 1–4 h exposure, the cells were preloaded with 20 μM DCFH-DA (Sigma-Aldrich, St. Louis, MO, USA) for 30 min and washed with PBS prior to lignan administration. After 24–72 h exposures, cells were washed once with PBS, loaded with DCFH-DA for 30 min, washed with PBS and incubated for further 3 h. After that, fluorescence was recorded at 503/523 nm with Varioskan Lux plate reader. The experiment was performed as four biological replicates.

### 4.10. Glutathione Quantification Assay

The intracellular GSH levels of THP-1 macrophages after *C. pneumoniae* infection, lignan treatment or combination of these two were determined using enzymatic recycling method described previously by Rahman et al. [51]. In brief, THP-1 monocytes were seeded into 6-well plate at density of 5 × 10^5^ cells/mL in cell culture medium without mercaptoethanol and differentiated into macrophages with PMA. The monolayers were exposed to either the lignan treatment (25 μM and 50 μM), *C. pneumoniae* infection (MOI5) or the combination of these two and incubated for a time indicated for each experiment. 0.25% DMSO were used as vehicle control. Cells were then collected, lysed and stored at −70 °C. To quantify GSH, glutathione reductase (GR) and 5,5’-dithiobis(2-nitrobenzoic acid) (DTNB) were added to the sample. After 30 s, β-NADPH were added and the formation of TNB chromophore, which is proportional to GSH concentration in the sample, was detected by multiskan Sky plate reader (Sigma-Aldrich). The concentrations of GSH were applied using linear regression to calculate the values obtained from the standard curves. 

For data normalization, total protein concentration determination of the cell lysates was performed with acetone precipitation. 100 μl of cell lysate sample was heated 5 min at 95 °C and 400 μl of cold (−20 °C) acetone was added. Sample was mixed and incubated 1 h at −20 °C, centrifuged at 15,000 *g* and supernatant was discarded. Pellet was resuspended to 100 mM Tris-buffer (pH; 7.5) and protein concentration was detected with Multiskan sky, µDrop plate. Sample purity was evaluated with 260/280 ratio values. The presented data are derived from two biological replicates both performed as two technical replicates (n = 4).

### 4.11. Data Analysis

Statistical tests were performed using SPSS Statistics 24 software. Differences between means were calculated with Student’s t-test with Bonferroni correction. *P*-values < 0.05 were considered statistically significant. Outliers were defined from data by Grubbs test, in significance level 0.05.

## 5. Conclusions

The current work demonstrates that the dibenzocyclooctadiene lignans schisandrin and schisandrin C possess antichlamydial activities that are qualitatively and quantitatively distinct from each other and the reference antibiotic azithromycin. The presented data also indicates that these activities are linked to the suppression of macrophage glutathione pools by the lignans. Redox status and glutathione homeostasis have recently emerged as modulators for intracellular bacteria virulence [46,52], indicating that targeting cellular redox mechanisms may offer a means for inducing a phenotypic switch in pathogenic bacteria. The superiority of schisandrin C compared to azithromycin in its ability to eliminate both active and persistent bacteria in THP-1 macrophages highlights the potential of this strategy in anti-persister therapy, triggering future research on nonconventional antibacterials. Furthermore, ability of schisandrin to resuscitate *C. pneumoniae* infection to the standard-of-care antibiotic azithromycin shows potential for further translational studies on these medicinal plant-derived compounds.

## Figures and Tables

**Figure 1 molecules-25-00294-f001:**
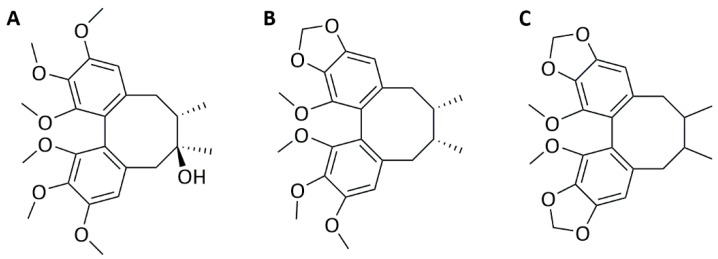
Chemical structures of dibenzocyclooctadiene lignans (**A**) schisandrin, (**B**) schisandrin B, and (**C**) schisandrin C.

**Figure 2 molecules-25-00294-f002:**
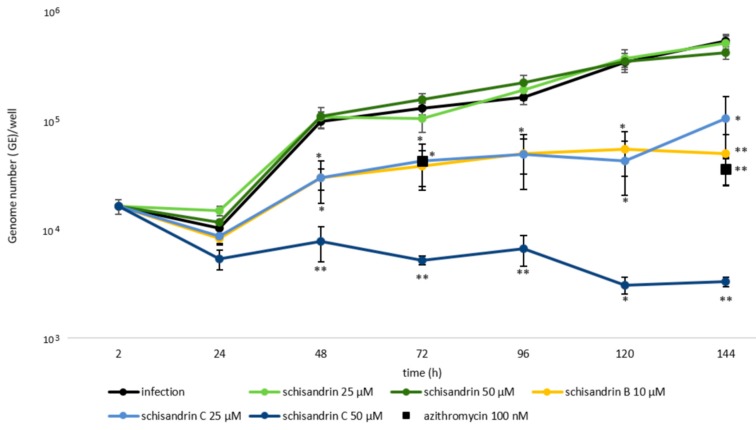
Impact of dibenzocyclooctadiene lignans on *C. pneumoniae* growth kinetics in THP-1 macrophages. Cells were infected at MOI 1 IFU/cell and *C. pneumoniae* genome copy numbers were determined with qPCR on ompA gene. Fresh medium was added at 72 h. Data are shown as total genome numbers of *C. pneumoniae* per well ± SEM (n = 4). Statistical significance is presented as marks of *p*-values: <0.05: *; <0.01: **. Abbreviations: MOI = multiplicity of infection.

**Figure 3 molecules-25-00294-f003:**
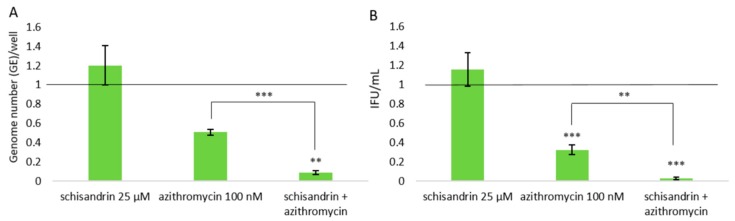
Impact of the concomitant administration of schisandrin (25 μM) and azithromycin (100 nM) on *C. pneumoniae* genome numbers (**A**) and EB progeny production (**B**). Data are normalized on the non-treated infection control and shown as a mean ± SEM (n ≥ 4) statistical significance is presented as marks of *p*-values: <0.01: **; <0.001: ***. Abbreviations: IFU = inclusion forming unit.

**Figure 4 molecules-25-00294-f004:**
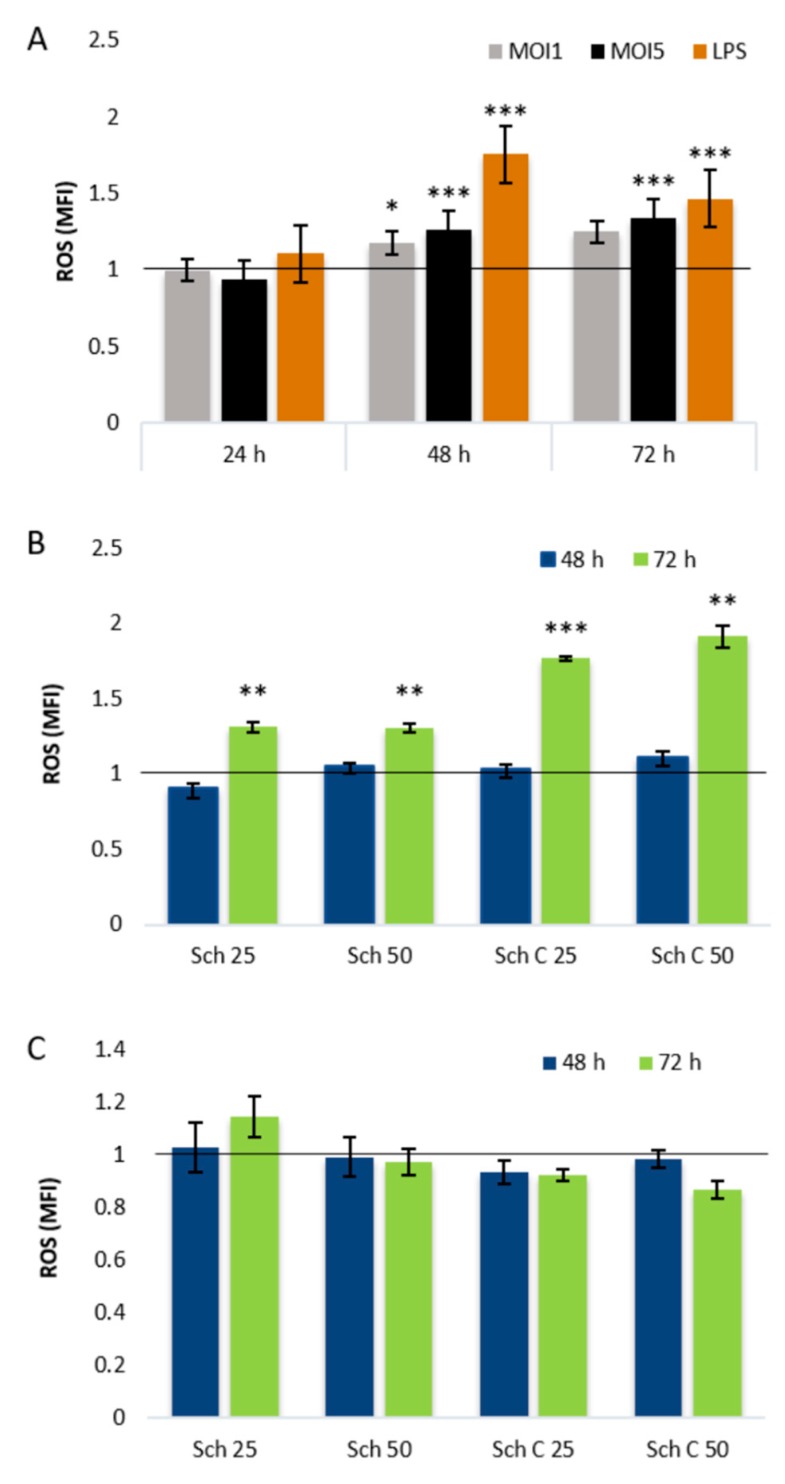
Impact of *C. pneumoniae* infection and dibenzocyclooctadiene lignans on intracellular ROS production. (**A**) THP-1 macrophages were infected with *C. pneumoniae* at MOI1 and MOI5 or treated with 1 μg/mL of LPS for 24, 48, or 72 h. (**B**) THP-1 macrophages were exposed to schisandrin lignans for 48 and 72 h. (**C**) THP-1 cells were simultaneously infected with *C. pneumoniae* and treated with schisandrin lignans for 48 and 72 h. Intracellular ROS levels were measured after 30 min incubation with DCFH-DA and followed by 3 h incubation with PBS. Data are normalized to a 0.25% DMSO vehicle cell control (**A**,**B**) or infection control (**C**) and shown as a mean ± SEM (n = 4) statistical significance is presented as marks of *p*-values: <0.05: *; <0.01: **; <0.001: ***. Abbreviations: Sch = schisandrin, MOI = multiplicity of infection.

**Figure 5 molecules-25-00294-f005:**
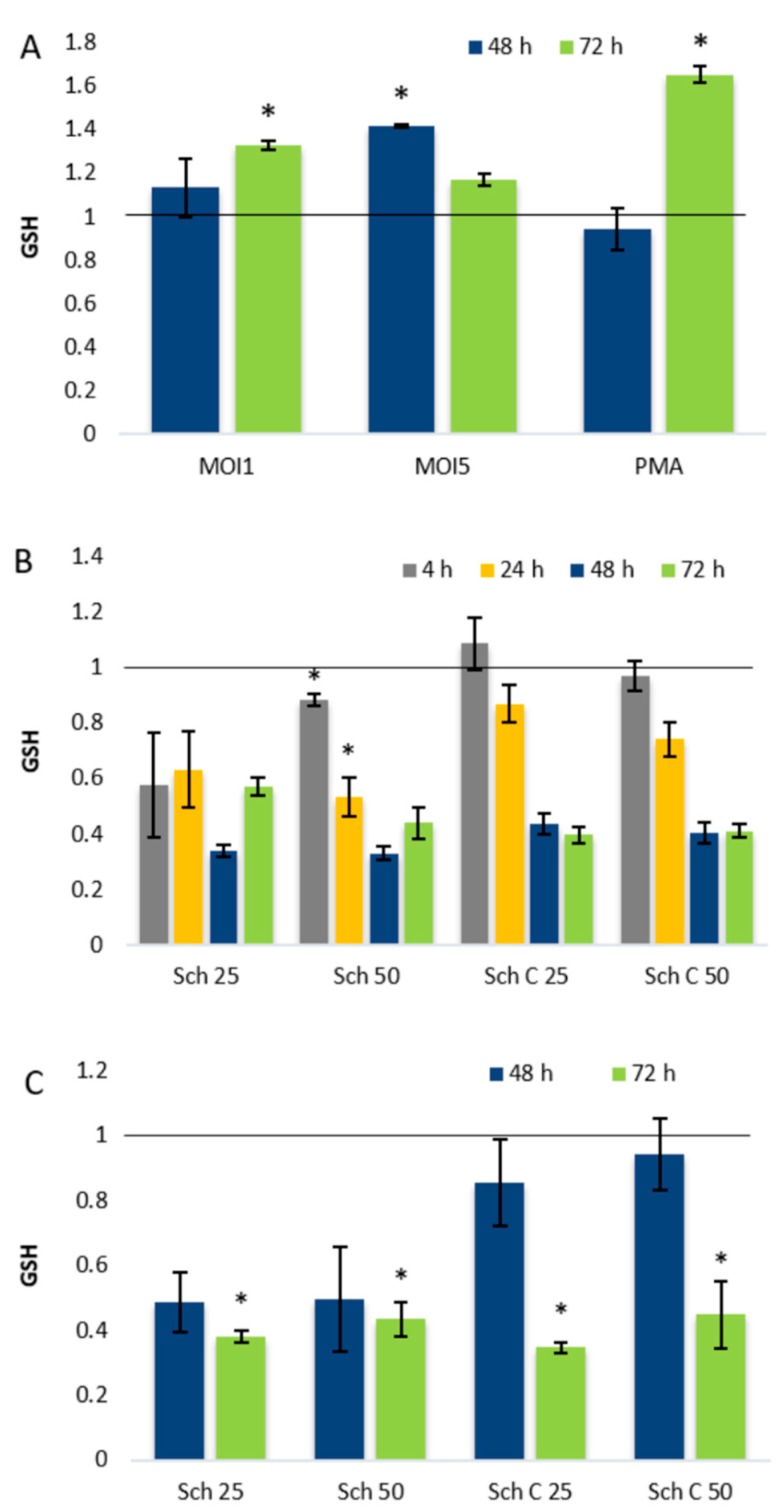
Effects of dibenzocyclooctadiene lignans on total cellular GSH levels in THP-1 macrophages. (**A**) THP-1 cells were infected with *C. pneumoniae* at MOI1 and MOI5 for 48 and 72 h. (**B**) THP-1 cells were exposed to schisandrin lignans for 4 to 72 h. (**C**) Cells were infected with MOI5 and exposed to schisandrin lignans for 48 and 72 h. Total GSH concentrations were determined and normalized to total protein concentration of the sample. Data are shown as a ratio of 0.25% DMSO control and shown as mean ± SEM (n = 4). Statistical significance is presented as marks of *p*-values: <0.05: *. Abbreviations: Sch = schisandrin, MOI = multiplicity of infection.

**Figure 6 molecules-25-00294-f006:**
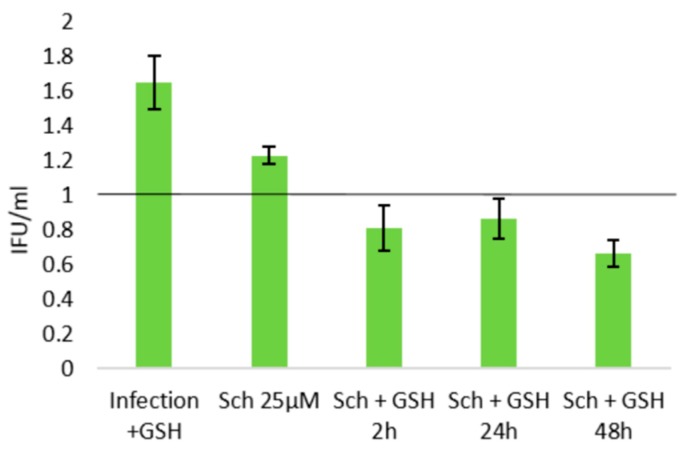
The impact of glutathione supplementation on schisandrin-induced changes in *C. pneumoniae* elementary body (EB) production. Data are normalized to the non-treated infection control and shown as a mean ± SEM (n = 4). Abbreviations: Sch = schisandrin.

**Table 1 molecules-25-00294-t001:** Impact of dibenzocyclooctadiene lignans on THP-1 macrophage cell viability with (Cpn +) and without (Cpn -) *C. pneumoniae* infection.

	24 h			48 h			72 h			144 h
Compound (µM)	Res	ATP		Res	ATP		Res	ATP		Res
	Cpn -	Cpn -	Cpn +	Cpn -	Cpn -	Cpn +	Cpn -	Cpn -	Cpn +	Cpn -
sch 25	118	113	120 *	105	172	122	93	103	99	127
sch 50	106	107	117	119	151	96	92	92	87	120
sch B 10	-	101	111	-	156	89	-	103	88	130
sch B 25	80	97	105	70	113	74	80 *	85	73	52 *
sch B 50	67 **	79 ***	87	50 **	92	57	69	53 ***	44 ***	1 **
sch C 25	89	94	100	78	139	96	111	102	90	122
sch C 50	79	90	108	69	121	81	101	93	82 *	81

Data are presented as viability percentages normalized to the values of 0.25% DMSO control. Statistical significance is presented as marks of *p*-values: <0.05: *; <0.01: **; <0.001: *** (n = 3). Abbreviations: Cpn *= C. pneumoniae*, sch = schisandrin, Res = resazurin.

**Table 2 molecules-25-00294-t002:** The Impact of dibenzocyclooctadiene lignans on *C. pneumoniae* infectious progeny production.

	Cell Lysate (IFU/mL Index)		Supernatant (IFU/mL Index)	
	72 h	144 h	72 h	144 h
sch 25 µM	1.53 ± 0.06 *	3.11 ± 0.62	1.52 ± 0.33	2.82 ± 0.27
sch 50 µM	2.28 ± 0.38	1.03 ± 0.34	2.56 ± 0.64	0.93 ± 0.18
sch B 10 µM	0.10 ± 0.03 **	0.01 ± 0.00 ***	0.28 ± 0.18	0.04 ± 0.01 ***
sch C 25 µM	0.00 ± 0.00 **	0.00 ± 0.00 ***	0.00 ± 0.00 **	0.00 ± 0.00 ***
sch C 50 µM	0.00 ± 0.00 **	0.00 ± 0.00 ***	0.00 ± 0.00 **	0.00 ± 0.00 ***
azithromycin 100 nM	0.18 ± 0.20 **	0.05 ± 0.03 ***	0.30 ± 0.21	0.00 ± 0.00***
infection (IFU/mL)	49,700 ± 16,000	140,200 ± 22,800	5400 ± 2100	12,300 ± 700

MOI 1 infection was used in experiments. Data are normalized to the non-treated infection control and presented as infectious units per mL (IFU/mL) ratios, showing as mean ± SEM. Results of non-treated samples are presented as IFU/mL ± SEM. < 0.05: *; < 0.01: **; < 0.001: ***, (n = 4). Abbreviations: sch = schisandrin, IFU = inclusion forming unit, MOI = multiplicity of infection.

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
