# Peer review of "Assaying Chlamydia pneumoniae Persistence in Monocyte-Derived Macrophages Identifies Dibenzocyclooctadiene Lignans as Phenotypic Switchers"

_molecules, 2020, doi:10.3390/molecules25020294_

Round 1

Reviewer 1 Report

This is an excellent paper providing novel insight into the mechanism of bacterial persistence, in this case of an intracellular pathogen, Chlamydia pneumoniae, using a class of natural products, lignans, as probes.  The paper is well written, the presentation is clear and compelling.  My only minor criticism is the lack of primary references regarding bacterial persistence in the introduction - since this is the topic of this paper.  I would suggest one or more of Phil Stewart's mechanistic papers (e.g. Borriello et al AAC,48(7)2659-2664, 2004) and one of  Jerry Aul's on the modern re-definition of persistence (e.g. Aul JJ, et al. Ann Otol Rhinol Laryngol. 1998 Jun;107(6):508-13.

Author Response

The authors wish to thank the reviewer for taking time for going through the work and are very grateful for the positive feedback. As a response, both articles suggested by the reviewer have been included in the references of the revised version.

Reviewer 2 Report

Dibenzocyclooctadiene lignans are medicinal plant-derived compounds which were used in this work against Chlamydia pneumonia inside THP-1 macrophages. They were shown to eliminate both active and persistent bacteria in THP-1. In anti-persister therapy , they resuscitated the dormant bacteria so that they became vulnerable to the standard-of-care antibiotic, azithromycin.

I found the work very interesting and important. Please, correct the following minor points:

1) Please, clarify the meaning of Cpn- and Cpn+ on the top caption describing the table.

2) On Figure 2, please, define GE (as the title of the y-axis).

3) Please, define “HL cells” first time they appear on the text.

4) Please, rephrase the last sentence of the Discussion. It does not make sense.

Author Response

The authors wish to thank the reviewer for taking the time to read the manuscript and are very grateful for the positive feedback. Below are listed the point-by-point responses to the comments:

1) Please, clarify the meaning of Cpn- and Cpn+ on the top caption describing the table.

Response: This has been clarified in the revised manuscript by including the explanation in the table caption.

2) On Figure 2, please, define GE (as the title of the y-axis).

Response: This has been corrected in the revised manuscript by writing out the abbreviation in the figure. For the sake of consistency, a similar change was made also to figure 3.

3) Please, define “HL cells” first time they appear on the text.

Response: This has been clarified in the revised manuscript by including the explanation ”Human Lung” into the text.

4) Please, rephrase the last sentence of the Discussion. It does not make sense.

Response: The end of the discussion section has been modified in the revised version in order to clarify the point.